# HyperMagNet: A Magnetic Laplacian based Hypergraph Neural Network

**Tatyana Benko**[*]                                                     *tbenko@uoregon.edu*
**Martin Buck**[*]                                                      *Martin.Buck@tufts.edu*
*University of Oregon*
*Tufts University*

**Ilya Amburg**                                                        *ilya.amburg@pnnl.gov*
*Pacific Northwest National Laboratory*

**Stephen J. Young**                                                   *stephen.young@pnnl.gov*
*Pacific Northwest National Laboratory*

**Sinan G. Aksoy**                                                     *sinan.aksoy@pnnl.gov*
*Pacific Northwest National Laboratory*

**Reviewed on OpenReview:** *https://openreview.net/forum?id=Gdf4P7sEzE*

## Abstract

In data science, hypergraphs are natural models for data exhibiting multi-way or group relationships in contrast to graphs which only model pairwise relationships. Nonetheless, many proposed hypergraph neural networks effectively reduce hypergraphs to undirected graphs via symmetrized matrix representations, potentially losing important multi-way or group information. We propose an alternative approach to hypergraph neural networks in which the hypergraph is represented as a non-reversible Markov chain. We use this Markov chain to construct a complex Hermitian Laplacian matrix — the magnetic Laplacian — which serves as the input to our proposed hypergraph neural network. We study *HyperMagNet* for the task of node classification, and demonstrate its effectiveness over graph-reduction based hypergraph neural networks.

## 1 Introduction

A fundamental limitation of graphs in machine learning is the relationships that a graph models are necessarily *pairwise*: edges connect exactly two vertices in a graph. For example, a graph edge may indicate two documents within a corpus share common vocabulary, two pixels in an image belong to the same neighborhood, or two diseases result from mutations in the same gene. However, these relationships are emphatically not only pairwise, but involve interactions between *groups* of documents, pixels, and diseases. Across these and other domains, modeling multi-way relationships with a graph therefore loses key information in the data. Hypergraphs, which allow more than two vertices to be connected by an edge, faithfully capture the multi-way relationships graphs cannot capture (Aksoy et al., 2020; Bretto, 2013; Feng et al., 2019). Data that is set-valued, tabular, or bipartite is best modeled with a hypergraph.

---

[*]Both authors contributed equally to this research. PNNL Information Release: PNNL-SA-194720. Please direct all inquiries about code availability to Sinan Aksoy.

Despite being a more expressive and general model than a graph, analyzing hypergraph data presents challenges. In machine learning tasks, one must first choose a *representation* of the hypergraph to be processed by a chosen algorithm. For example, convolutional neural networks use a Hermitian matrix representation of the hypergraph in order to learn optimal node embeddings via matrix multiplication with learnable weight matrices. However, classic matrix representations of graphs such as the adjacency matrix or the graph Laplacian have no direct analogues for hypergraphs. To overcome this challenge, representations of a hypergraph via a *random walk* have proved convenient. Zhou et al. (2006) demonstrated this, defining a clustering algorithm based on a reversible random walk, where there is symmetry in the transition probabilities between any two vertices, and the eigenvectors of an associated hypergraph Laplacian. Building on this representation and inspired by the success of graph convolutional neural networks (GCN), the seminal hypergraph neural network HGNN (Feng et al., 2019) uses Zhou's hypergraph Laplacian in a traditional GCN defined by Kipf & Welling (2016) for classification tasks in visual object recognition and citation network classification.

Unfortunately, it is known (Agarwal et al., 2006) that Zhou's hypergraph Laplacian indeed reduces to a graph Laplacian on the star graph corresponding to a hypergraph, and other Laplacians reduce to those of the clique graph. In this sense, these Laplacians and the aforementioned hypergraph neural network HGNN reduce a hypergraph to a graph. The reason this information loss occurs is because these matrices are based on reversible random walks, which always reduce to a random walk on an undirected graph (Chitra & Raphael, 2019). Thus, a more faithful approach is to define Laplacians based on non-reversible random walks. As shown by Chitra & Raphael (2019), hypergraph random walks which utilize *edge-dependent vertex weights (EDVW)*, in which transition probabilities are guided by vertex-hyperedge specific weightings, may be non-reversible. These weights, which allow vertices to have varying importance across the hyperedges to which they belong, often naturally present in data or may also be derived from structural properties of unweighted hypergraph data.

In this work, we build and study a hypergraph neural network that doesn't rely on a star graph or clique expansion reduction Laplacian. Rather, our proposed *HyperMagNet* begins with a non-reversible hypergraph random walk that is more faithful in capturing nuances within hypergraph data, which are reflected as asymmetries in transition probabilities. However, the transition probability matrix representing this random walk is not Hermitian which complicates their use within traditional convolutional neural networks. Instead of overcoming this by only symmetrizing the transition matrix (thereby converting to a reversible, graph-based random walk), we instead encode it as a complex-valued, Hermitian-but-asymmetric matrix called the magnetic Laplacian. This Laplacian has been successfully applied in the graph learning community to directed graphs (Fanuel et al., 2017; Zhang et al., 2021; Shubin, 1994). Studying its application in a hypergraph setting, we explain how to learn parameters of the magnetic Laplacian from hypergraph data, and build a hypergraph neural network around it. Finally, we investigate the efficacy of this approach, in comparison to HGNN and related graph-reduction methods for the task of node classification. On varied data, we find that *HyperMagNet* outperforms competing graph-based methods, sometimes modestly and sometimes significantly, and include experiments to test whether this increase in performance is due to the utilization of edge-dependent vertex weights, the magnetic Laplacian, or both. Although slightly more expensive to run, *HyperMagNet* is worth using due to its increase in performance over graph-reduction based models, as shown across several data modalities.

The contributions of this paper are outlined as follows:

- We propose a new hypergraph neural network, *HyperMagNet*, that doesn't rely on a graph representation of the hypergraph. Instead, *HyperMagNet* uses a non-reversible hypergraph random walk that is more faithful in capturing nuances within hypergraph data.

- We explain how to learn parameters of the magnetic Laplacian, a complex-valued, Hermitian-but-asymmetric matrix, and build a hypergraph neural network around it.

- We investigate the efficacy of *HyperMagNet* in comparison to HGNN and related graph-reduction methods for the task of node classification.

The paper is structured as follows: in Section 2, we provide background on hypergraphs, hypergraph random walks, and the magnetic Laplacian. In Section 3 we show how the magnetic Laplacian is an appropriate

hypergraph Laplacian for the EDVW random walk and include comparisons with traditional hypergraph random walks and Laplacians based on clique graphs. We also introduce the neural network architecture of *HyperMagNet* in this section. Section 4 contains related work on hypergraph neural networks with and without EDVW. In Section 5 are experimental results in the tasks of node classification on a variety of hypergraph structured data sets, where performance is compared against a variety of machine learning models based on traditional graph-based representations.

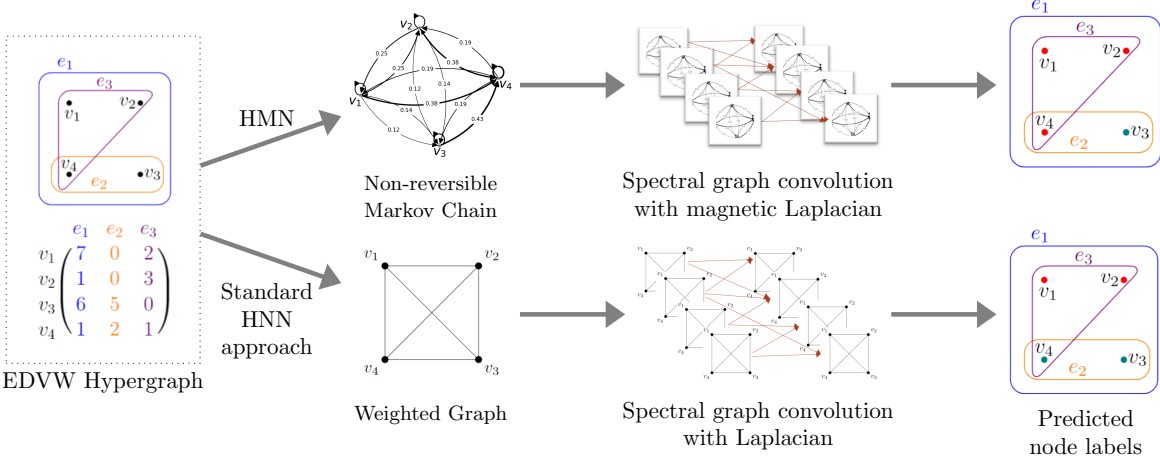

Figure 1: *HyperMagNet* (HMN) uses a non-reversible Markov chain to build a hypergraph Laplacian which avoids Laplacians associated with the star graph or clique expansion.

## 2 Background

### 2.1 Hypergraphs and Random Walks.

A *hypergraph* $H = (V, E)$ is a set of vertices $V = \{v_1, \ldots, v_n\}$ and hyperedges $E = (e_1, \ldots, e_m)$ where each $e_i \subset V$ for $i = 1, 2, \ldots, m$. A graph is a special case of a hypergraph where all hyperedges have size 2. The vertex-hyperedge relationship is given by the *incidence matrix*, $Y \in \{0, 1\}^{|V| \times |E|}$, where $y(v, e)$ is 1 if $v \in e$ and is 0 otherwise.

A simple approach to analyze a hypergraph is to represent it as a graph in a way that retains important information from the hypergraph, thereby enabling the application of graph machine learning methods. One popular such graph representation is the *clique expansion* which replaces hyperedges by sets of edges forming cliques. This graph has weighted adjacency matrix $YY^T$, where the weights correspond to the number of shared hyperedges between two vertices. Another popular graph representation of a hypergraph is the *star expansion* $G^* = (V^*, E^*)$ which introduces a new vertex for each hyperedge $e \in E$ so that $V^* = V \cup E$ and $E^* = \{(v, e) : v \in e, e \in E\}$.

Unsurprisingly, graph representations of hypergraphs may lose information. A simple example is when many small hyperedges are contained within a larger hyperedge; the clique expansion will erase the relationships between vertices in the sub-hyperedges. Furthermore, non-isomorphic hypergraphs may have identical clique and line graphs, as we illustrate later in Figure 2b. However, working with hypergraph-to-graph reductions is convenient because it enables application of graph learning methods. Graph random walks and Laplacians are used throughout machine learning, such as in recommendation systems, spectral clustering, and *graph convolutional neural networks (GCN)*. In order to develop similar tools for hypergraphs, a spectral theory is necessary. Here, Zhou et al. (2006) proposed a spectral clustering algorithm for hypergraphs based on a hyperedge partitioning problem rooted in a simple hypergraph random walk. Letting $\omega(e) > 0$ denote the *hyperedge weight* associated with $e$, $d(v) = \sum_{\{e \in E | v \in e\}} \omega(e)$ denote the vertex degree of $v$, and $|e| = \delta(e)$ denote the *hyperedge degree*, the random walker in this construction at time $t$ and vertex $v_t$ proceeds by:

1. Choose a hyperedge $e \ni v_t$ with probability proportional to hyperedge weight $\omega(e)$

2. Select vertex $v \in e$ uniformly at random

3. Move to vertex $v_{t+1} := v$ at time $t+1$

The transition matrix $P$ for this random walk has the form $p(v, u) = \sum_{e \in E} \omega(e) \frac{y(v,e)}{d(v)} \frac{y(u,e)}{\delta(e)}$. Letting $D_V$, $D_E$, and $W$ denote the diagonal vertex degree, hyperedge degree, and hyperedge weight matrices, respectively, $P$ is $D_V^{-1} Y W D_E^{-1} Y^T$. Zhou et al. then define a *hypergraph Laplacian* as:

$$\Delta = I - D_V^{-1/2} Y W D_E^{-1} Y^T D_V^{-1/2} \tag{1}$$

This process of selecting a vertex uniformly at random is an example of an *edge-independent vertex weighting (EIVW)*. For a hyperedge weighting function $\gamma_e : V \to \mathbb{R}_+$, if $\gamma_e(v) = \gamma_{e'}(v)$ for all pairs $e, e'$ containing $v$, then the collection $\{\gamma_e\}_{e \in E}$ is an edge-independent vertex weighting. Otherwise if the $\gamma_e(v)$ varies across hyperedges, the vertex weighting is referred to as *edge-dependent vertex weighting (EDVW)*. EDVW appear across applications: in NLP where the weights are tf-idf values representing the importance of a word to a document (Bellaachia & Al-Dhelaan, 2013); in e-commerce, where weights correspond to the number of items in a shopper's basket (Li et al., 2018); or in biology, where the weights correspond to association scores between a gene and a disease (Feng et al., 2021). In the absence of weight data, EDVW may also be generated from hypergraph structure, as explored later in Section 5.2.

## 2.2 The Representative Digraph of a Hypergraph.

Chitra & Raphael (2019) show any hypergraph random walk using EIVW is equivalent to a random walk on the clique graph. Consequently, hypergraph learning methods that use EIVW to construct a hypergraph random walk or hypergraph Laplacian may lose higher-order relationships in the data. To remedy this, Chitra & Raphael (2019) show EDVW random walks are not necessarily equivalent to a random walk on any undirected graph such as the clique expansion. Inspired by Hayashi et al. (2020), we use the EDVW to represent the hypergraph via an *EDVW random walk* on the hypergraph where, if at vertex $v_t$ at time $t$, the next step is chosen as follows:

1. Select a hyperedge $e \ni v_t$ with probability proportional to hyperedge weight $\omega(e)$

2. Select a vertex $v \in e$ with *probability proportional to EDVW $\gamma_e(v)$*

3. Move to vertex $v_{t+1} := v$ at time $t+1$

A key difference between the EDVW random walk and that defined in Zhou et al. (2006) is that the former may be *non-reversible* (Chitra & Raphael, 2019), while the latter is always reversible. Reversibility means that, in stationarity, the probability of transitioning from $i$ to $j$ is the same as from $j$ to $i$ (more formally, $\pi_i P_{ij} = \pi_j P_{ji}$, where $\pi$ is the stationary distribution and $P$ is the transition matrix). Reversible random walks are equivalent to random walks on graphs, whereas non-reversible random walks are not. Hence, the EDVW random walks cannot be characterized as random walks on the hypergraph's clique graph. This EDVW random walk generalizes the simple random walk defined by Zhou et al. (2006) since we allow for a larger class of probability distributions in the second step above, rather than limiting to a uniform distribution. The EDVW information is stored in a weighted incidence matrix, $R$, where $R_{ve}$ is $\gamma_e(v)$ if $v \in e$ and is 0 otherwise. The transition matrix $P$ for this EDVW random walk is given by $p(v, u) = \sum_{e \in E} \frac{\omega(e)}{d(v)} \frac{\gamma_e(u)}{\delta(e)}$ or $P = D_V^{-1} Y W D_E^{-1} R^T$ in matrix notation. We represent $P$ as a directed graph called the *representative digraph* of the hypergraph (Hayashi et al., 2020), with vertex set $V$ and weighted edge set $E = \{(u, v) \mid P_{uv} > 0\}$. As this random walk may be non-reversible, special tools from spectral graph theory are necessary to construct a spectral-based neural network and define graph convolutions (e.g., Defferrard et al. (2016); Kipf & Welling (2016)) that rely on the spectral theorem.

We stress EDVW random walks – which are key to our magnetic Laplacian approach – can provably capture higher-order structure otherwise lost by graph approaches. We provide two examples that illustrate this in

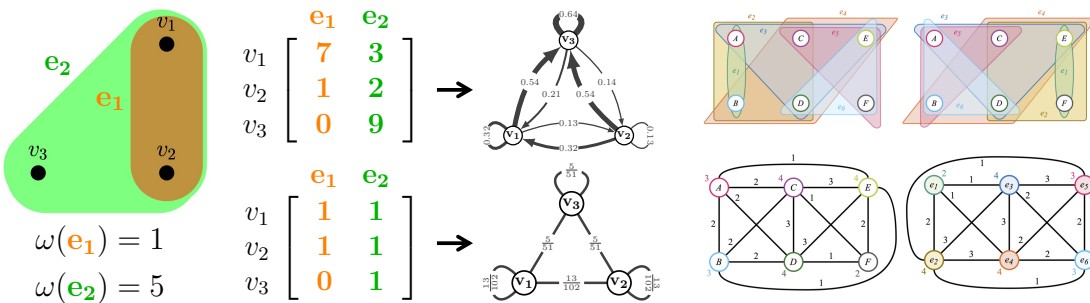

(a) Edge dependent vs. independent vertex weightings and the resulting reversible and non-reversible Markov chains

(b) Non-isomorphic hypergraphs with identical weighted clique and line graphs

Figure 2: *Left*: EDVW (top) vs EIVW (bottom) random walks yielding graph vs digraph random walks. *Right*: hypergraphs (top) with identical, intersection-weighted clique and line graphs (bottom) yet are distinguished by their EDVW random walk properties and magnetic Laplacian spectra.

two senses. Figure 2a illustrates how EIVW hypergraph random walks may collapse to a graph random walk, whereas factoring in EDVW instead may avoid this. Observing the differences between the representative digraphs also provide intuition about the different weightings: the transition probabilities from $v_3$ to $v_1$ or $v_2$ are equal in the EIVW case, whereas the large EDVW weight $\gamma_{e_1}(v_1) = 7$ yields a larger probability of transitioning to $v_1$ than $v_2$. Figure 2b is an example from Aksoy et al. (2024) of subtly-different hypergraphs with the same underlying graph information (as given by identical weighted clique and line graph expansions), yet are distinguished by EDVW random walk properties. A witness to their non-isomorphism is that an alternating vertex-hyperedge walk, the diameter of hypergraph $S$ is 5, whereas for $R$ it is 4. Using the structure-derived EDVW weighting $\gamma_j(i) = \deg(v_i)/\sum_{v_k \in e_j} \deg(v_k)$, the maximum expected hitting time in an EDVW random walk on $S$ is about 12.24 between nodes $A$ and $F$, whereas this hitting time is 11.41 in hypergraph $R$. It can also be verified the magnetic Laplacian spectra differ across the two examples, which we define next.

## 2.3   The Magnetic Laplacian.

A remarkable tool for analyzing digraphs, such as this representative digraph of the hypergraph, is the *magnetic Laplacian*. The magnetic Laplacian is built on a complex-valued Hermitian adjacency matrix and has its origins in the quantum physics literature as the Hamiltonian of a charged particle confined to a lattice under the influence of magnetic forces. An obvious difference between the magnetic Laplacian and the standard array of graph Laplacians is that direction information appears as a signal in the complex-plane. However, it still enjoys the properties that convolutional neural networks are built on such as being Hermitian and positive semi-definite.

Previous work using a complex-valued Hermitian adjacency matrix or Laplacian can be found in Cucuringu et al. (2019), Fanuel et al. (2018), Zhang et al. (2021), Fiorini et al. (2022). In Cucuringu et al. (2019) the eigenvectors of a complex-valued Hermitian matrix are used to cluster migration networks in the United States, revealing long-distance migration patterns that traditional clustering methods miss. Similarly, in Fanuel et al. (2018) the spectrum of the magnetic Laplacian is shown to be related to solutions of the angular-synchronization problem akin to how the spectrum of the graph Laplacian is related to solutions of the graph cut problem. The authors also successfully use the eigenvectors to cluster word-adjacency and political blog networks. Zhang et al. (2021) use the magnetic Laplacian to define a new graph convolution and build a neural network for directed graphs that beats state-of-the-art directed graph learning algorithms on a number of NLP data sets. There is also recent work on incorporating the magnetic Laplacian into graph neural networks that process signed and directed graphs, see Fiorini et al. (2022), He et al. (2022). In both Fiorini et al. (2022) and He et al. (2022), a novel Laplacian derived from the magnetic Laplacian is introduced, specifically designed to handle signed and directed graphs in a spectral GCN. In

Fiorini et al. (2022), the authors propose the sign-Magnetic Laplacian, which is parameter-free, unlike the magnetic Laplacian. In contrast, He et al. (2022) present the signed magnetic Laplacian which retains a single charge parameter.

The magnetic Laplacian is defined as follows. Let $A_s$ be the *symmetrized adjacency matrix* $A_s := \frac{1}{2}(A + A^T)$, and let $D_s$ denote the corresponding diagonal degree matrix. Now, define $\Theta^{(q)}(A)$ as the skew-symmetric *phase matrix* $\Theta^{(q)}(A) := 2\pi q(A - A^T)$. For a given parameter $q \geq 0$, the complex-valued Hermitian adjacency matrix $H^{(q)}(A)$ is given by an entrywise product between the symmetrized adjacency matrix and an entrywise exponential of the phase matrix:

$$H^{(q)}(A) := A_s \odot \exp(i\Theta^{(q)}) \tag{2}$$

The *unnormalized magnetic Laplacian* is $D_s - H^{(q)}$ while the *normalized magnetic Laplacian* is

$$L^{(q)}(A) := I - D_s^{-1/2} A_s D_s^{-1/2} \odot \exp(i\Theta^{(q)}) \tag{3}$$

The parameter $q \geq 0$ is called a *charge parameter* and determines how direction information is represented and processed. When $q = 0$, the phase matrix $\Theta^{(0)}(A) = 0$ and so $H^{(0)}(A) = A_s$, thereby removing direction information in the graph. When $q \neq 0$, $H^{(q)}(A)$ may be complex-valued, whose imaginary components capture direction information through the $A - A^T$ term in the phase matrix. As shown in Fanuel et al. (2018), varying $q$ changes how patterns and motifs in the graph are captured via the spectrum. Choosing an optimal value of $q$ a priori is a difficult task and previous work has treated $q$ as a hyperparameter. Since we consider weighted digraphs, a single value of $q$ may be insufficient to capture direction information. To better accommodate different edge weightings when training *HyperMagNet*, we introduce a novel *charge matrix Q* (given later in Eq. 6 and 7) of learnable parameters that replace the single charge parameter $q$ in the Laplacian. As far as the authors know, this is the first work to analyze a hypergraph via the EDVW random walk and spectrum of a magnetic Laplacian which opens up applications within hypergraph signal processing by mimicking approaches to clustering, de-noising, and visualization as seen in Cucuringu et al. (2019) and Fanuel et al. (2018), for example.

## 3 HyperMagNet

### 3.1 A Hypergraph Laplacian and Convolution.

We define a *hypergraph Laplacian* using the non-reversible Markov chain captured in the representative digraph $P$,

$$L^{(q)}(P) := I - D_s^{-1/2} P_s D_s^{-1/2} \odot \exp(i\Theta^{(q)}) \tag{4}$$

To simplify notation, the hypergraph Laplacian $L^{(q)}(P)$ will be written as $L^{(q)}$. Since $L^{(q)}$ is a positive semi-definite matrix, by the *spectral theorem*, we have an eigendecomposition of the hypergraph Laplacian $L^{(q)} = \Phi\Lambda\Phi$, where $\Phi$ is the matrix whose columns are its eigenvectors $\{\phi_1, \ldots, \phi_n\}$, and $\Lambda$ is the diagonal matrix containing its corresponding non-negative eigenvalues, $\{\lambda_1, \ldots, \lambda_n\}$. A hypergraph *Fourier transform* of a function $x \in \mathbb{C}^n$ on the vertices of the hypergraph is then defined as the representation of the signal in terms of the hypergraph Laplacian eigenbasis, $\hat{x} = \Phi^* x$. This mimics how a graph Fourier transform was defined in Shuman et al. (2016) and for directed graphs in Zhang et al. (2021). Furthermore, because the matrix $\Phi$ is unitary, we can express the original function in terms of its hypergraph Fourier transform: $x = \Phi\hat{x} = \Phi(\Phi^* x) = \sum_{k=1}^{N} \hat{x}(k)\phi_k$. In classical Fourier analysis, a convolution of two functions $f : \mathbb{R}^n \to \mathbb{C}$ and $g : \mathbb{R}^n \to \mathbb{C}$ has the key property that convolution corresponds to multiplication of the Fourier transforms, $\widehat{f * g} = \hat{f}\hat{g}$. Following Shuman et al. (2016) and Zhang et al. (2021), we define a *hypergraph convolution* as pointwise multiplication of two functions $x \in \mathbb{C}^n$, $y \in \mathbb{C}^n$ on the vertices of the hypergraph in the eigenbasis:

$$\widehat{y * x}(k) = \hat{y}(k)\hat{x}(k) \tag{5}$$

This can be expressed in matrix notation as $y * x = \Phi\text{Diag}(\hat{y})\Phi^* x$. Therefore, for a fixed function $y$ its corresponding convolution matrix is $C_y = \Phi\text{Diag}(\hat{y})\Phi^*$. Then, convolution with $x$ can be expressed as the matrix-vector product $y * x = C_y x$. Following the construction of a convolution in Kipf & Welling (2016)

and Feng et al. (2019), we approximate convolution in Eq. 5 by a truncated Chebyshev polynomial and renormalize the hypergraph Laplacian to have eigenvalues in the range $[-1, 1]$. This circumvents the cost of computing an eigenbasis and performing the Fourier and inverse Fourier transform. The resulting hypergraph convolution has the form $C_y x = \theta_0 \left( I + \tilde{D}_s^{-1/2} \tilde{P}_s \tilde{D}_s^{-1/2} \odot \exp(i\Theta^{(q)}) \right) x$, where $\theta_0$ is a learnable parameter, $\tilde{D}_s$ and $\tilde{P}_s$ are the diagonal degree matrix and adjacency matrix corresponding to the re-normalized Laplacian $\tilde{L}^{(q)} := \frac{2}{\lambda_{\max}} L^{(q)} - I$.

### 3.2 A Learnable Charge Matrix and HyperMagNet Architecture.

We outline *HyperMagNet's* network architecture and introduce a modification to the magnetic Laplacian with a learnable charge matrix $Q$ to accommodate the weighted directed edges present in the hypergraph's representative digraph. Prior work Zhang et al. (2021); Fanuel et al. (2018); Cucuringu et al. (2019) studying the magnetic Laplacian and its applications in clustering and neural networks assume the underlying digraph is unweighted, which helps inform the choice of the single charge parameter $q$, typically in $0 \le q \le .25$. When $q \ne 0$, the phase encodes edge direction and the Hermitian adjacency matrix $H_{uv}^{(q)}(A)$ takes on four values: $H_{uv}^{(q)}(A) = 0, \exp(2\pi i q), \exp(-2\pi i q), 1$ if, respectively: there are no edges between $u$ and $v$, only from $u$ to $v$, only from $v$ to $u$, and reciprocal edges. When $q = 1/4$, if there is an edge from $u$ to $v$ but not from $v$ to $u$, then $H_{uv}^{(1/4)}(A) = \frac{i}{2} = -H_{vu}^{(1/4)}(A)$. In this case, an edge from $v$ to $u$ is treated as the "opposite" of an edge from $v$ to $u$ (Zhang et al., 2021). Mohar (2020) asserts a natural choice is $q = \frac{1}{3}$ as $H_{uv}^{(1/3)}(A) = e^{\pi i/3}$ is now a sixth root of unity and thus two oppositely oriented edges have the same effect as a single undirected edge. Nonetheless, it is not clear a priori which choice of $q \ge 0$ is optimal.

For the representative digraph of a hypergraph, the edges are weighted and the justifications for a universal $q$ break down. Since $\Theta_{uv}^{(q)}(P) = 2\pi q[P_{uv} - P_{vu}]$, there are large phase angles when there is a high probability of moving from $v$ to $u$ but not $u$ to $v$, i.e. when the transition matrix is more asymmetric. All that is accomplished by setting $q = 1/4$ or $q = 1/3$ is restricting the phase angle between $-\pi/2$ and $\pi/2$ or $-2\pi/3$ and $2\pi/3$, respectively. This means the effects of a single charge parameter $q$ are not sufficient for the entire graph if we are to mimic the effect of setting $q$ to a fixed value. *Therefore $q$ should vary with edge weight in order to accomplish what a single $q$ accomplishes in the unweighted case.* As there is no optimal choice of $q$ a priori, we introduce a *charge matrix $Q \in \mathbb{R}^{n \times n}$* of learnable charge parameters. The Hermitian adjacency matrix now has the form

$$H^{(Q)}(P) = P_s \odot \exp(i\Theta^{(Q)}), \tag{6}$$

where $\Theta^{(Q)}(P) := 2\pi i Q \odot (P - P^T)$. The *weighted hypergraph Laplacian* then has the same form as before except replacing $H^{(q)}$ with $H^{(Q)}$:

$$L^{(Q)}(P) := I - D_s^{-1/2} P_s D_s^{-1/2} \odot \exp(i\Theta^{(Q)}) \tag{7}$$

To simplify notation, the weighted hypergraph Laplacian used in *HyperMagNet* $L^{(Q)}(P)$ is denoted $L^{(Q)}$ and the renormalized version $\tilde{L}^{(Q)}$. Like with GCNs, each layer of the network will transform the previous layer's vertex feature matrix via matrix multiplication with a matrix of learnable parameters that correspond to the filter weights $\theta_0$ in Eq. 5. We let $L$ be the number of layers in the network and let $X^{(0)}$ be the the $n \times f_0$ vertex feature matrix. Here, $n$ corresponds to the number of vertices in the hypergraph and $f_0$ is the length of a feature vector associated with each vertex. For $1 \le l \le L$, let $f_l$ be the number of channels or hidden units in the $l$-th layer of the network. In matrix notation, if $W_{\text{self}}^{(l)}$ and $W_{\text{neigh}}^{(l)}$ are the learnable filter weight matrices, and $Q$ is the learnable charge matrix, the output of the $l$-th layer is then:

$$X^{(l)} = \sigma(X^{(l-1)} W_{\text{self}}^{(l)} + \tilde{L}^{(Q)} X^{(l-1)} W_{\text{neigh}}^{(l)} + B^{(l)}) \tag{8}$$

where $B^{(l)}$ is a matrix of real bias weights with the form $B^{(l)}(v, \cdot) = (b_1^{(l)}, \ldots, b_{f_l}^{(l)})$, for $v \in V$. Since $\tilde{L}^{(Q)}$ is complex-valued, the activation function $\sigma$ is a complex-valued ReLU which zeroes out input in the left half of the complex plane and defined as $\sigma(z) = z$ if $Arg(z) \in [-\pi/2, \pi/2]$, and $\sigma(z) = 0$ otherwise. After the $L$ convolutional layers, the real and imaginary parts of the complex-valued node feature matrix $X^{(L)} \in \mathbb{C}^{n \times f_L}$ are separated and then concatenated into a real-valued feature matrix $\hat{X}^{(L)} \in \mathbb{R}^{n \times 2f_L}$. Finally, we apply a

linear layer via multiplication with learnable weight matrix $W^{(L+1)} \in \mathbb{R}^{2f_L \times n_c}$ mapping the learned node features to a vector corresponding to the number of classes $n_c$. This is then transformed into a vector of class probabilities via softmax.

## 4 Related Work

### 4.1 Hypergraph Neural Networks.

Since the seminal work of Feng et al. (2019) introducing HGNN, several other spectral-based hypergraph neural networks (HNN) have been proposed. HyperGCN (Yadati et al., 2019) applies a GCN to a weighted graph representation of an EIVW hypergraph and uses a non-linear Laplacian (Chan & Liang, 2020; Louis, 2015) to process the graph structure in the convolutional layers. Zhang et al. (2022) introduce a unified random walk hypergraph Laplacian which can be used in a GCN for both EDVW and EIVW hypergraphs. Their approach, however, incorporates the EDVW information via a reversible random walk, and thus the hypergraph data is still represented by a weighted graph. Other approaches such as Hayhoe et al. (2024) also use a spectral based approach, but in the context of quantifying similarity between hypergraphs via graph expansions such as the clique expansion and line graph.

### 4.2 Edge-Dependent Vertex Weights in Hypergraph Neural Networks.

Zhang et al. (2022) build the equivalency between EDVW hypergraphs and undirected graphs via a unified random walk. This random walk on the hypergraph incorporates two different EDVW matrices, $Q_1$ and $Q_2$, one for each step of the random walk, whose probability transition matrix is given by

$$P = D_V^{-1} Q_1 W \rho(D_E) Q_2^T \tag{9}$$

where $\rho$ is a function of the diagonal hyperedge weight matrix $D_E$. They show if both $Q_1$ and $Q_2$ are edge-independent, or $Q_1 = kQ_2$ for some $k \in \mathbb{R}$, then there exist weights on the hypergraph's clique expansion such that the random walk given by Eq. 9 is equivalent. They use this equivalency to build a hypergraph neural network on existing graph convolutional neural networks through this graph representation. The non-reversible EDVW random walk (Chitra & Raphael, 2019) used in *HyperMagNet* can be obtained from Eq. 9 with $Q_1 = Y$ (the incidence matrix) and $Q_2 = R$ (the EDVW matrix), and thus does not satisfy either of the aforementioned conditions for equivalency to a graph random walk.

## 5 Experiments

To incorporate EDVW into HGNN, we use the EDVW random walk given by Eq. 9. We use $\rho(X) = X^{-1}$ and $Q_1 = Q_2 = R$, the EDVW matrix. We use this reversible EDVW random walk in place of the simple random walk used in Zhou's hypergraph Laplacian, defined in Eq. 1, i.e.

$$\Delta = I - D_V^{-1/2} R W D_E^{-1} R^T D_V^{-1/2} \tag{10}$$

We use HGNN* to denote HGNN using this Laplacian. While this incorporates EDVW into HGNN, the underlying random walk is still reversible and thus loses higher-order hypergraph structure.

### 5.1 Term-Document Data.

The 20 Newsgroups data set consists of approximately 18,000 message-board documents categorized according to topic. To test the performance of *HyperMagNet (HMN)* on predicting which topic a document belongs to, subsets of four categories were chosen as in Hayashi et al. (2020). The first set of four categories (G1) includes documents in the OS Microsoft Windows, automobiles, cryptography, and politics-guns topics. The second set (G2) consists of documents on atheism, computer graphics, medicine, and Christianity. The third (G3) contains documents on Windows X, motorcycles, space, and religion. Finally, the last set (G4) contains documents on computer graphics, OS Microsoft Windows, IBM PC hardware, MAC hardware, and Windows X. The categories in G4 are expected to be similar presenting a more difficult classification problem.

For all subsets of categories, the documents undergo standard cleaning by removing headers, footers, quotes, and pruned by removing words that occur in more than 20% of documents using PorterStemmer. Following Hayashi et al. (2020), Chitra & Raphael (2019), the hypergraph is created with documents as vertices, words as hyperedges, tf-idf values as the EDVW, and hyperedge weights as the standard deviation of the EDVW of vertices within each hyperedge. Tf-idf is a common weighting scheme for tokens in a document. Whereas bag-of-words uses a binary representation for a token indicating the presence or absence of a token in a document, tf-idf is a scalar representation that incorporates frequency information of the word within a document weighted agaits its frequency across documents in the corpus. The formula is a term (token) frequency multiplied by an inverse document frequency, as seen below. Let $t$ represent a token in document $d$ in the corpus of documents $D$. Then,

$$\text{tf-idf}(t, d) = \left( \frac{f_{t,d}}{\sum_{t' \in d} f_{t',d}} \right) * \left( \log \frac{|D|}{d : d \in D \text{ and } t \in d} \right).$$

The interpretation is that terms that occur frequently within a particular document but are rare in other documents should be given greater weight than those that also occur frequently across all documents. Table 1 (left) shows the resulting hypergraph sizes.

The average classification accuracy over ten random 80%/20% train-test splits for each of the subsets G1 through G4 is recorded in Table 1. Both HGNN ands *HyperMagNet* are two layer neural networks that follow standard hyperparameter settings based on that in Kipf & Welling (2016). The dimension of hidden layers set to 128 with ReLU activation functions. For training the Adam optimizer was used to minimize the cross-entropy loss a the learning rate of 0.001 and weight decay at 0.0005. These settings were used for training across data sets. The same settings are used in Kipf and Welling's GCN for a baseline comparison across experiments. For another spectral-based graph comparison, we run spectral clustering on the clique expansion and use a majority-vote on the clusters to assign labels.

The following experiments feature two options for a Laplacian in HGNN: (1) the standard hypergraph Laplacian using the unweighted hypergraph incidence matrix as defined by Zhou et al. (2006) and seen in Eq. 1 (this is the Laplacian that HGNN architecture is built on); and (2) our weighted hypergraph Laplacian defined in Eq. 10 using the weighted hypergraph incidence matrix which incorporates EDVW information. We also include experimental results where the feature vectors $X^{(0)}$ are the standard bag-of-words representation of a document, and the tf-idf values. Table 1 shows *HyperMagNet* outperforms the graph-based models by a large margin across all categories with the exception of when EDVW information is used in the Laplacian for HGNN, where the advantage is smaller.

Table 1: *Left*: Hypergraph sizes for 20 Newsgroups data in chosen subsets G1 through G4. *Right*: Average node classification accuracy on 20 Newsgroups. Best result for each subset G1-G4 is **bold**, second best is underlined. *HyperMagNet* (HMN) is highlighted in yellow. HGNN and HyperGCN are other spectral hypergraph methods, HGNN* is HGNN with the EDVW incorporated as given in Eq. 10. GCN is a standard graph convolutional network and GCN* uses weighted edges from tf-idf information in the Laplacian. Spectral clustering is performed on the clique expansion with majority-vote labeling. Node features are tf-idf values or bag-of-words (BoW) representation.

|  | Nodes | Hyperedges |
|---|---|---|
| G1 | 2,243 | 13,031 |
| G2 | 2,204 | 9,351 |
| G3 | 2,110 | 9,766 |
| G4 | 2,861 | 12,938 |

|  | G1 | G2 | G3 | G4 |
|---|---|---|---|---|
| HMN (tf-idf) | **90.33**±1.15 | **90.6**±0.95 | 92.66±0.78 | **78**±1.69 |
| HMN (BoW) | 89.2 ±0.93 | 89.61±0.94 | 92.44±0.69 | 77.87±1.02 |
| HGNN (tf-idf) | 69.34±2.09 | 69.97±1.42 | 75.69±1.82 | 40.9±2.11 |
| HGNN (BoW) | 79.34±2.01 | 79.74±2.80 | 88.44±1.23 | 59.67±2.31 |
| HGNN* (tf-idf) | 77.08±2.12 | 76.55±1.65 | 90.36±1.02 | 66.71±2.75 |
| HGNN* (BoW) | 89.39±0.95 | 89.71±0.92 | **92.85**±0.91 | 76.28±1.58 |
| HyperGCN (tf-idf) | 78.30±1.57 | 75.99±2.87 | 89.36±1.25 | 68.08±2.12 |
| HyperGCN (BoW) | 89.58±1.88 | 89.90±1.12 | 91.82±1.20 | 76.49±2.16 |
| GCN* (tf-idf) | 48.11±2.45 | 48.39±2.05 | 52.08±3.09 | 51.97±2.67 |
| GCN* (BoW) | 49.82±2.01 | 50.2±3.06 | 52.43±3.75 | 50.88±4.04 |
| GCN (tf-idf) | 31.89±4.77 | 30.01±2.94 | 34.51±5.75 | 52.87±2.40 |
| GCN (BoW) | 32.84±5.89 | 32.34±4.60 | 32.38±5.21 | 52.3±2.69 |
| Spec. Clustering | 51.13±1.17 | 59.21±2.04 | 61.13±2.01 | 47.04±1.95 |

## 5.2 Citation and Author-Paper Networks.

The Cora Citation data set (McCallum et al., 2000) consists of citations between machine learning papers classified into seven categories of topics with the BoW representation for each paper. The hypergraph is constructed with papers as vertices and citations as hyperedges. One hyperedge per paper is generated, and this hyperedge contains the paper and all papers it cited. Not all papers cited others in the data set; these degree one hyperedges are not included in the hypergraph. We also run *HyperMagNet* on the Cora Author (McCallum et al., 2000) data set. This hypergraph has papers as vertices and authors as hyperedges; an author's hyperedge contains papers that author appeared on.

Both Cora Citation and Cora Author do not have the raw text available to construct the EDVW from tf-idf values. Therefore, we define the EDVW matrix $R$ based on the degree of the vertices within each hyperedge as

$$R_{ij} = \frac{\deg(v_i)}{\sum_{v \in e_j} \deg(v)} \tag{11}$$

This may be a natural EDVW in many contexts: more weight is assigned to vertices with larger degree reflecting greater importance to a particular hyperedge. Interpreted in the context of citation networks, this choice of EDVW measures the "citation-prestige" of a paper $i$ relative to that of all papers cited by hyperedge $j$. For example, if $j$ mostly cites niche or obscure research along with one highly-cited impact paper $i$, the EDVW of paper $i$ with respect to $j$ will be high. In contrast, if $i$ is cited in another paper $k$ which cites many highly-cited impact papers, the EDVW for $(i, k)$ will be lower. We note this choice of EDVW takes one particular view of "relative citation prestige", which (by design) biases the RW towards highly cited articles. Furthermore, it only uses *local* information to define citation prestige (i.e. the citation count of $i$ and that of its neighbors within $j$). Using a more nuanced centrality measure beyond degree, like PageRank, to define $R_{ij}$ would yield a more complex notion of citation prestige. This range of expressivity in EDVW is a strength of *HyperMagNet* approach.

Since *HyperMagNet* can be flexibly run with or without EDVW information, we include results that incorporate both the above hypergraph degree EDVW and a simple EIVW instead. Table 2 shows *HyperMagNet* outperforms HGNN on Cora Author by a margin of 4%, whereas HGNN slightly outperforms *HyperMagNet* on Cora Citation.

Table 2: *Left*: Hypergraph sizes for Cora Author and Cora Citation. *Right*: Average node classification accuracy (%) on the Cora Author and Cora Citation data sets. Best result is **bold**, second best is underlined. *HyperMagNet* (HMN), with EDVW and EIVW, is highlighted in yellow. HGNN and HyperGCN are other spectral hypergraph methods, HGNN* is HGNN with the EDVW incorporated as given in Eq. 10. GCN is a standard graph convolutional network. Spectral clustering is performed on the clique expansion with majority-vote labeling.

|  | Nodes | Hyperedges |
|---|---|---|
| Cora Author | 2,388 | 1,072 |
| Cora Citation | 1,565 | 2,222 |

|  | Cora Author | Cora Citation |
|---|---|---|
| HMN | **88.01±0.97** | 85.62±1.28 |
| HMN EIVW | 87.87±1.06 | 85.73±1.21 |
| HGNN | 83.11±0.45 | 83.82±1.20 |
| HGNN* | 83.01±0.32 | **85.94±1.51** |
| HyperGCN | 83.62±0.88 | 85.10±1.04 |
| GCN | 75.89±1.50 | 76.81±1.69 |
| Spec. Clustering | 84.92±1.92 | 81.66±2.11 |

## 5.3 Computer Vision.

Princeton ModelNet40 (Wu et al., 2015) and the National Taiwan University (NTU) (Chen et al., 2003) are two popular data sets within the computer vision community to test object classification. The ModelNet40 data set is composed of 12,311 objects from 40 different categories. The NTU data set consists of 2,012 3D shapes from 67 categories. Within each data set, each object is represented by features that are extracted using two standard shape representation methods called Multi-view Convolutional Neural Network (MV) and Group-View Convolutional Neural Network (GV).

Following the construction in Feng et al. (2019), the hypergraph is created by grouping vertices together into a hyperedge that are within a nearest-neighbor distance based on the above features in the data. That is, a probability graph based on the distance between vertices is constructed using the RBF kernel $W_{ij} = \exp{-2D_{ij}/\Delta}$ and the nearest ten neighbors for each vertex are grouped together in a hyperedge. The NTU hypergraph has 2,012 nodes and hyperedges, while the ModelNet40 hypergraph has 12,311 nodes and hyperedges. The EDVW are the $W_{ij}$ values so that the close vertices are given more weight than distant vertices. Here, *HyperMagNet* outperforms the other models in the range of $1 - 7\%$ depending on the Laplacian and feature vector combination.

Table 3: Average node classification accuracy (%) on NTU and ModelNet40 data sets. Best result is **bold**, second best is underlined. *HyperMagNet* (HMN) is highlighted in yellow. HGNN and HyperGCN are other spectral hypergraph methods, HGNN* is HGNN with the EDVW incorporated as given in Eq. 10. GCN is a standard graph convolutional network. Spectral clustering is performed on the clique expansion with majority-vote labeling. Node features GV and MV are extracted using two standard object representation methods.

|  | | GV for hypergraph construction | | MV for hypergraph construction | |
|---|---|---|---|---|---|
|  | Node fts | NTU | ModelNet40 | NTU | ModelNet40 |
| HMN | GV | **94.93** $\pm$ 0.91 | 91.76 $\pm$ 0.57 | **94.4** $\pm$ 0.88 | **98.73** $\pm$ 0.15 |
| HMN | MV | 94.87 $\pm$ 0.90 | **98.46** $\pm$ 0.20 | 91.21 $\pm$ 1.37 | 97.22 $\pm$ 0.31 |
| HGNN | GV | 93.48 $\pm$ 0.98 | 89.82 $\pm$ 0.40 | 87.30 $\pm$ 1.33 | 91.44 $\pm$ 0.54 |
| HGNN | MV | 93.24 $\pm$ 1.13 | 97.22 $\pm$ 0.31 | 86.98 $\pm$ 1.43 | 97.14 $\pm$ 0.26 |
| HGNN* | GV | 93.21 $\pm$ 1.27 | 90.55 $\pm$ 0.58 | 87.84 $\pm$ 1.85 | 92.20 $\pm$ 1.01 |
| HGNN* | MV | 94.80 $\pm$ 0.92 | 95.74 $\pm$ 0.89 | 88.40 $\pm$ 1.09 | 96.15 $\pm$ 0.65 |
| GCN | GV | 41.8 $\pm$ 4.95 | 69.46 $\pm$ 6.23 | 32.25 $\pm$ 5.53 | 79.05 $\pm$ 5.72 |
| GCN | MV | 55.27 $\pm$ 6.96 | 80.97 $\pm$ 15.6 | 47.44 $\pm$ 3.54 | 95.51 $\pm$ 1.15 |
| HyperGCN | GV | 92.53 $\pm$ 0.88 | 90.65 $\pm$ 1.21 | 88.86 $\pm$ 0.97 | 92.27 $\pm$ 0.97 |
| HyperGCN | MV | 94.81 $\pm$ 1.23 | 96.02 $\pm$ 1.09 | 88.56 $\pm$ 1.17 | 96.13 $\pm$ 1.32 |
| Spec. clustering | - | 75.55 $\pm$ 2.34 | 91.80 $\pm$ 3.32 | 63.5 $\pm$ 3.08 | 91.95 $\pm$ 4.17 |

### 5.4 Ablation Study.

Finally, we compare the performance of *HyperMagNet* using the weighted hypergraph Laplacian (Eq. 7) and the unweighted hypergraph Laplacian (Eq. 4). The purpose here is to demonstrate the utility of introducing the charge matrix $Q$ in the Laplacian as a replacement for the charge parameter $q$. This was motivated in Section 3 as a method to account for the varying edge weights and directionality in the representative digraph that could not be accounted for by previous work with the magnetic Laplacian. Table 4 shows on the Cora Citation, Cora Author, and 20 Newsgroups data sets that there is clear performance gain with the weighted hypergraph Laplacian.

Table 4: Performance comparison between *HyperMagNet* (HMN) using the $L^{(Q)}$ and $L^{(q)}$ Laplacians defined in Eq. 7 and Eq. 4, average node classification accuracy (%) on Cora Author, Citation, and 20 Newsgroups (G4).

|  | Cora Author | Cora Citation | 20 Newsgroups (G4) |
|---|---|---|---|
| HMN $L^{(Q)}$ | **85.62**$\pm$1.63 | **88.01**$\pm$0.97 | **78**$\pm$1.69 |
| HMN $L^{(q)}$, $q = 0$ | 83.71$\pm$1.71 | 81.44$\pm$1.52 | 75.70$\pm$2.14 |
| HMN $L^{(q)}$, $q = .15$ | 83.98$\pm$1.14 | 81.47$\pm$1.55 | 75.98$\pm$2.24 |
| HMN $L^{(q)}$, $q = .25$ | 83.62$\pm$1.31 | 81.60$\pm$1.70 | 75.65$\pm$2.05 |

### 5.5 Runtime Comparison and Scalability Discussion.

The additional computational cost of *HyperMagNet* can be attributed to three factors: (1) complex-valued arithmetic (2) the introduction of the learnable charge matrix and (3) dense blocks within the probability transition matrix $P$. A complex-valued neural network like *HyperMagNet* incurs the cost of doing complex-

valued arithmetic. Matrix multiplication is still naively a cubic-order operation, but each multiplication of two complex numbers is four times the number of operations as the multiplication of two real numbers. The charge matrix $Q$ adds an additional $O(n^2)$ (where $n$ is the number non-zero entries in the transition matrix $P$) operations to each forward pass in the network. This is due to the Hadamard (entry-wise) matrix product that is introduced via the new phase matrix $\Theta^{(Q)}(P) = 2\pi i Q \odot (P - P^T)$ that is absent in the original phase matrix $\Theta^{(q)} = 2\pi i q(P - P^T)$. This effect is ameliorated when $P$ is sparse, which unfortunately is not always the case in practice. For example, in the 20 Newsgroups G4 dataset there is an order of magnitude more hyperedges than nodes in the hypergraph and large hyperedge sizes $k$. This contributes to a dense transition matrix $P$ and the order of magnitude jump in training time of *HyperMagNet* compared to other models. A possible barrier to scaling *HyperMagNet* – and more generally, all of the hypergraph methods surveyed here – is that large hyperedges yield dense matrices. Indeed, any hypergraph random walk in which there is always a nonzero probability of transitioning between vertices within the same hyperedge yields $O(k^2)$ nonzeros in $P$ and subsequently $L^{(Q)}$. Several techniques can be applied to ameliorate this issue. One could take a hyperedge filtering approach, applying one of the filtering operations defined and studied in Landry et al. (2024). In contexts where filtering out large hyperedges is undesirable, an alternative approach is to retain them but construct a sparser transition matrix via hypergraph sparsification algorithms Liu et al. (2022); Benson et al. (2020). For example, Algorithm 2 in Liu et al. (2022) is an efficient and parallelizable approach for computing the $s$-line graph or $s$-clique expansion of a hypergraph. Interpreted in our context, this may be adapted to restrict transition space to pairs of vertices which share $s$ or more hyperedges, yielding sparser transition matrices as $s$ is increased.

Table 5: Model runtime comparison reported in minutes.

| | Cora Author | Cora Citation | 20 Newsgroups (G4) |
|---|---|---|---|
| HMN $L^{(Q)}$ | 0.71 | 0.55 | 43.7 |
| HMN $L^{(q)}$, $q = 0$ | 0.49 | 0.37 | 40.6 |
| HMN $L^{(q)}$, $q = .15$ | 0.57 | 0.44 | 41.4 |
| HMN $L^{(q)}$, $q = .25$ | 0.48 | 0.37 | 40.9 |
| HGNN | 0.16 | 0.43 | 3.62 |
| HyperGCN | 0.21 | 0.54 | 3.10 |
| GCN | 0.14 | 0.34 | 2.92 |
| Spec. Clustering | 0.11 | 0.44 | 3.64 |

### 5.6 Future Work.

In addition to improving runtime and boosting *HyperMagNet's* capacity to process large hypergraphs by incorporating sparsification methods, we have identified two other potential areas for future work: link prediction and directed hypergraphs. While performing link prediction on hypergraphs is feasible, and hypergraph neural networks have been proposed for this task Yadati et al. (2020), several challenges are associated with it. First, predicting the existence of hyperedges is more complex for hypergraphs than for standard graphs, as hyperedges can involve any number of vertices. Second, while *HyperMagNet* could be used to learn node and hyperedge embeddings from the input hypergraph, a scoring function would still be necessary to evaluate the probability of existence for both existing and potential hyperedges. This would require defining and integrating such a function within the HMN framework.

Incorporating directed hypergraphs into *HyperMagNet* is a task that can be more easily achieved. Given a random walk on a directed hypergraph with EDVW, it can be represented using the corresponding representative digraph and processed with the magnetic Laplacian in *HyperMagNet*. Random walks on directed hypergraphs have been explored, see Ducournau & Bretto (2014). However, there are important considerations when defining the random walk for use in *HyperMagNet*. Depending on the dataset and the random walk definition (such as transitioning from a vertex in the head of a hyperedge to a vertex in the tail), the walk may not be ergodic. To address this, we can incorporate a restart mechanism into the random walk to ensure its ergodicity.

## 6 Conclusion

We proposed *HyperMagNet*, a hypergraph neural network which represents the hypergraph as a non-reversible Markov chain, and uses the magnetic Laplacian to process the higher-order information for node classification tasks. In building this Laplacian, we integrated a learnable charge matrix which allows *HyperMagNet* to better process the weights associated with the non-reversible Markov chain. We demonstrated the performance of *HyperMagNet* against other spectral-based HNNs and GCNs on several real-world data sets. Our results suggest the more nuanced and faithful approach taken by *HyperMagNet* may lead to performance gains, ranging from modest to significant, for the task of node classification. Further investigation of *HyperMagNet* is warranted, including adapting it to perform other tasks such as link prediction, incorporating directed hypergraphs (Gallo et al., 1993) as possible inputs to the model, and using hypergraph sparsification methods (Liu et al., 2022) to improve training time.

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
