# OpenReview forum: "HyperMagNet: A Magnetic Laplacian based Hypergraph Neural Network"
_TMLR — Accepted by TMLR_

### Review · Reviewer_6krg · 2024-12-12

**Summary Of Contributions:**

The authors propose a novel hypergraph neural network that is based on non-reversible Markov chains, allowing for a more detailed distinction between different hypergraphs. The authors show better performance to other methods and evaluate their method on several datasets.

**Audience:**

Yes

**Claims And Evidence:**

Yes

**Requested Changes:**

- Adapting the presentation of the results (see above) to make them more clear and highlight the specific contribution as well as important findings
- Potentially, re-doing the experiments with the same train/test splits.
- The authors note that the runtime increases with their method. It would be good to add those to the experiments to get an idea of the magnitude.
- The conclusion could be extended to highlight main findings more clearly and show the limitations of this work. Here, also a discussion on the runtime could be added, when the numbers are evaluated in the results section.
- How easy would it be to extend this method to other methods such as link prediction? The authors mention briefly that this is part of future investigation. Here it would be good to discuss a bit how this would be possible.
- It would be interesting to discuss in which settings your method out-performs others? Do you know in which settings HMN might be especially suitable to use? why is the performance difference larger for some datasets and smaller in others? Is this linked to the nature of the hyperedges or graph size or sth like that?

**Strengths And Weaknesses:**

Strengths:
- The approach sounds very reasonable, really using the full hypergraph structure
- The methods are explained well and it is well written
- Nice ablation study comparing weighted and unweighted Laplacian

Weaknesses:
- The presentation of the results could use some more clarity in my opinion. This includes the tables e.g. I think that all tables should be readable without having to check in the text for abbreviations. Maybe add explanations to the caption. Also it would be good to highlight clearly which method is yours and which are related works.
- What does the asterisk mean in Table 1?
- It is not clear to me why the authors opted for using 10 random splits on the dataset instead of evaluating different randomly initialised models. Could you please elaborate on this?
- As far as I've seen there is no code base linked. This would help assessing the work and make it useable for others
- Were hyperparameters tuned individually for all datasets and methods? Because the same parameters could favour specific methods.
- The authors only evaluate accuracy in their experiments. It would be interesting to see if the method has an impact on other metrics and scores, such as F1 or similar.

Minor writing/formatting issues:
- “Observe the differences between the representative digraphs also provide intuition about the different weightings:” —> “Observing the differences…” (page 4)
- The legends of Figure 2 b and 2a (most right) are too small for reading. This should be improved
- “However, it still enjoys the properties that a convolutional neural networks are built on such as being Hermitian and positive semi-definite.” —> “However, it still enjoys the properties that convolutional neural networks…” (page 4)
- Repetition: there are no edges between u and v, only from u to v, only from v to u, and reciprocal edges.

---

> ### Author Response · Authors · 2025-02-25
> **Reviewer 6krg responses**
>
> We'd like to thank the reviewer for carefully reading the paper and helpful comments/suggestions for improvement.
>
> * We appreciate your suggestion to enhance the clarity and readability of our results presentation, as well as to highlight our specific contributions. We have highlighted the results of HyperMagNet in each table to clearly distinguish our method. Additionally, we have provided explanatory details in the table captions to ensure they are easily understood without the need to reference the text for abbreviation clarification. As for highlighting our specific contributions, we have added a bulleted list of our specific contributions in the Introduction.
>
> * Regarding the reviewer's comment on re-doing the experiments with the same train/test splits, we note that we did randomly initialize model parameters for each experiment, and followed standard practice to average across train/test splits. We emphasize that the same train/test splits were used across all models for a fair comparison. We hope that clarifies the referee’s concern, please let us know if we can provide more information.
>
> * Thank you for the suggestion to add runtime data. We have included a section prior to the conclusion that discusses the complexity and runtime scalability of HyperMagNet, along with potential sparsification methods to be explored as future work aimed at enhancing the practicality of HyperMagNet for large hypergraphs.
>
> * We have added a discussion on the potential application of HyperMagNet to link prediction and the incorporation of directed hypergraphs in a section titled "Future Work" prior to the Conclusion.
>
> * While we don’t expect HyperMagNet to outperform other models in all scenarios, we agree explaining and predicting when this will be the case is a very interesting question. This is a challenging question to prove definitively, though the directionality factored in by HMN leads one to conjecture that when the hypergaph EDVW random walk is more “asymmetric” as reflected in some measure of asymmetry in the representative digraph, HyperMagNet will generally outperform graph-reduced models in the vein of HGNN. Measures of asymmetry could be Li and Zhang’s digraph singular value, the “principal ratio” (of maximum to minimum values in the stationary distribution), or something more crude such as the average entry difference between a matrix and its transpose. In our initial set of experiments with a single charge parameter q, we saw weak correlations between measures of asymmetry and the performance difference between HyperMagNet and the other models, but did not include them due to space limitations and the weakness of the correlation. However, a direction for future work we’d be very interested in exploring is understanding what the correct measure of “asymmetry” is here and using HyperMagNet with the charge matrix Q.
>
> * HGNN* in Table 1 and the other tables indicates that that the HGNN model was used with EDVW incorporated. This explanation has been added to the caption of all tables. A detailed explanation of how EDVWs were integrated into HGNN can be found in the first paragraph of Section 5.
>
> * Regarding open sourcing our code -- we are eager to do so! However, due to the employment contract of certain authors (which we won’t specify to respect the double-blind process) any kind of external software release must undergo certain institutional approval processes that take a significant amount of time and are out of our control. We ask the referees if the following would be acceptable to address their concerns about reproducibility:
>
>     * We will include a note in the manuscript saying that the “the code may be made available upon personal request to the authors, pending institutional approval”.
>
>     * We are happy to further specify, either as an appendix and/or within the main text, any additional details to ensure repoducibility. We remind the referees that our architecture is described in Section 3, and our hyperparameter choices are defined at the beginning of Section 5. However, if the referees have other specific details they’d also like us to include here, please let us know!
>
> * The hyperparameters were fixed by default according to the recommendations of the authors of the original papers. For all models (where applicable), the hyperparameters were set to the same values. This is discussed in the third paragraph on page 8.
>
> * We explored a number of metrics, such as the F1 score, Jacard Index, and Mutual Information to evaluate performance and found no differences in the performance rankings. In fact, the F1 score completely mimicked accuracy in rankings. This is not surprising, as the classes are relatively balanced and we wouldn’t expect that incorporating recall and precision information to change the rankings.
>
> * Thank you for noticing the minor writing/formating errors, we have fixed them.

---

### Review · Reviewer_pa67 · 2024-12-13

**Summary Of Contributions:**

The authors begin by explaining how existing approaches for analyzing hypergraphs often involve converting the hypergraph into a standard graph, followed by the application of spectral techniques on the resulting graph. They then highlight the limitations of this process and introduce an alternative method: directly applying spectral techniques to hypergraphs using the magnetic Laplacian. Their proposed method, HyperMagNet, demonstrates promising results in node classification tasks across various hypergraph domains.

As someone who is not an expert in hypergraphs, I recognize that some of my comments may seem basic. That said, I recommend clearly defining certain concepts (even if they might seem straightforward to experts) to make the paper more accessible and engaging for readers without a deep background in hypergraphs. Considering my non-expertise in hypergraphs, the following comments are an educated guess.

**Audience:**

Yes

**Broader Impact Concerns:**

no ethical concerns

**Claims And Evidence:**

Yes

**Requested Changes:**

1. Better introduce the notion of non-reversible random walk
2. Better explain the difference concerning Fiorini et al 2022 and He et  al. 2022
3. Not clear why table 1 contains 4 GCN items
4. Details on the difference between (tf-idf) and (BoW)
5. Citation order:  (Aksoy et al., 2020; Bretto, 2013; Feng et al., 2019) -->  (Aksoy et al., 2020; Feng et al., 2019; Bretto, 2013).

**Strengths And Weaknesses:**

**Strengths**

The paper is well-written and effectively motivates the proposed approach, making the ideas easy to follow and understand.
The integration of edge-dependent vertex weights (EDVW) adds significant flexibility, enabling the model to handle diverse hypergraph structures and making it applicable to various real-world scenarios.

**Weaknesses**

The proposed method is computationally intensive compared to traditional graph-reduction approaches. This increased cost may limit its practicality for processing very large hypergraphs.

Some aspects of the experimental setup are not clearly explained, which could hinder reproducibility and understanding (details provided below).

---

> ### Author Response · Authors · 2025-02-25
> **Reviewer pa67 responses**
>
> We thank the reviewer for their careful reading, review, and feedback on the paper.
>
> * Concerns with the increased cost is an important point, and we thank the reviewer for the observation. In response, we have included a section prior to the conclusion that discusses the complexity and runtime scalability of HyperMagNet, along with potential sparsification methods to be explored as future work aimed at enhancing the practicality of HyperMagNet for large hypergraphs.
>
> * We agree it is worth formally defining the notion of a non-reversible random walk given the importance of this concept to our approach. We now include a formal definition in the background section. We do note the critical takeaway about reversibility is mentioned early on in the introduction (“The reason this information loss occurs is because these matrices are based on reversible random walks, which always reduce to a random walk on an undirected graph”).
>
> * We have incorporated additional sentences where Fiorini et al. (2022) and He et al. (2022) were previously cited, providing a comparison of the similarities and differences between the two magnetic Laplacian based neural networks.
>
> * Thank you for pointing out the additional rows of GCN results. Two of these rows should be labeled with a '\*' because they also use tfidf information in their Laplacian, as with HGNN and HGNN*.
>
> * We have added clarification on the difference between the tf-idf and BoW representations of a token in Section 5.1.
>
> * We are using alphabetical ordering for the citations. If the journal has a different requirement we are happy to change it.

---

### Review · Reviewer_J8KU · 2025-02-10

**Summary Of Contributions:**

The manscript introduces HyperMagNet (HMN), a hypergraph neural network architecture focused on node classification. HMN employs the magnetic Laplacian, constructed from a non-reversible Markov chain based on edge-dependent vertex weights (EDVW), to represent hyper-graph structure.

The core contribution is the introduction of a learnable charge matrix, replacing the traditional fixed charge parameter within the magnetic Laplacian. This seems to address the limitation of single charge parameters in capturing complex weighted directed edges in hypergraph representative digraphs. HMN, also, utilizes a complex-valued neural network with a complex ReLU activation function. The author(s) show an empirical evaluation across diverse datasets (term-document, citation networks, computer vision) that demonstrates that HMN outperforms existing hypergraph and graph neural networks, showcasing its ability to better capture multi-way relationships in hypergraphs compared to graph-based reduction methods.

**Audience:**

Yes

**Broader Impact Concerns:**

I do not believe there exist of any concerns for ethical implications  that would require  adding a Broader Impact Statement.

**Claims And Evidence:**

Yes

**Requested Changes:**

The most important changes I'd recommend to be integrated are the following:

Computational Complexity Discussion: Include a paragraph discussing computational complexity, comparing HMN to real-valued counterparts and including runtime scaling analysis.

Ablation Study Extension (Scalar Charge): Add an ablation experiment comparing HMN to a version with a learnable scalar charge parameter to isolate the benefit of the charge matrix.

Reproducibility: It'd be very useful if the authors provide some form of resource (e.g. anonymous repo) to check for reproducibility.

Moreover, it'd be positive if the authors can discuss the following :

Theoretical Properties of Charge Matrix: Expand discussion on theoretical properties of the learned charge matrix Q, exploring convergence, interpretability, and regularization.

EDVW for Citation Networks: Elaborate on the justification and potential limitations of degree-based EDVW for citation networks.

Hyperparameter Sensitivity: Include a short paragraph discussing hyperparameter sensitivity and provide practical tuning guidance, especially for the charge matrix.

**Strengths And Weaknesses:**

**Strengths**

Novelty and Significance: the authros introduce a learnable charge matrix in magnetic Laplacian for HGNNs, which is novel and well-justified, addressing limitations of fixed charges. To my knowledge, its tje first work to combine EDVW random walks and magnetic Laplacians for hypergraph learning.

Strong Empirical Validation: The authors use diverse datasets to evaluate their method showing competitive and often superior performance compared to HGNN, HyperGCN, GCN, and spectral clustering. HGNN* comparison highlights magnetic Laplacian benefits.

Clear Motivation: Effectively motivates non-reversible random walks and magnetic Laplacian. Clearly explains limitations of Zhou et al.'s Laplacian. Strong theoretical grounding with EDVW, representative digraphs and quantum mechanics analogy. Figure 2 effectively illustrates key concepts.

Well-Written and Organized: Clear background, method, and experiment sections. Consistent mathematical notation and effective placement within existing literature.

**Weaknesses:**

Reproducibility : Although the authors give a fair description of the architecture, the experimental setup, and the data preprocessing steps, the lack of code, or an anonymous repo hinders me from understanding whether the experiments are reproducible.

Computational Complexity: As far as I understand, Complex-valued networks and learnable charge matrix introduce a computational overhead. Needs discussion of computational complexity of HyperMagNet vs. real-valued HGNNs/GCNs, including runtime scaling analysis.

Limited Theoretical Charge Matrix Analysis: Limited theoretical exploration of the learnable charge matrix's properties (convergence, interpretability, regularization).
Ablation Scope (Scalar Charge): In order to realize matrix form benefits, a version with a learnable scalar charge could be compared with the HMN model.

EDVW Derivation Heuristics: Justification and limitations of degree-based EDVW derivation for citation networks could be expanded.
Hyperparameter Sensitivity: Due to the lack of the code, I cannot identify any sensitivity of hyperparameters. The authors should discuss any sensitivity observations they can yield.

---

> ### Author Response · Authors · 2025-02-25
> **Reviewer K8KU responses**
>
> We thank the reviewer for their careful reading and suggestions for the paper.
>
> * It is a good idea to discuss the computational complexity of HMN compared to other models. We have added a new section “Runtime Comparison and Scalability Discussion”, where we compare the runtimes across the different hypergraph NNs, and discuss the computational complexity. We also include a discussion of the space complexity challenges posed by HyperMagNet and related approaches, and recommend several approaches for ameliorating this issue.
>
> * We thank the reviewer for the suggestion to extend the ablation study. Indeed, we haven’t run experiments where 'q' is a learnable parameter, we’ve only run experiments where it is treated as a hyperparameter. If we treated a single 'q' as a learnable parameter, this doesn’t solve the problem introduced by the edge weights: it still applies the same across all edges and in particular edges of different weights. This is why we introduced the charge matrix and didn’t perform experiments further with a single charge parameter. In the ablation study seen in Table 4, we experiment with a range of parameter values, q=0,0.15, 0.25, (as it is set in previous works with the magnetic Laplacian) which we believe covers a sufficient range to isolate the effects and benefits of the charge matrix.
>
> * Regarding open sourcing our code -- we are eager to do so! However, due to the employment contract of certain authors (which we won’t specify to respect the double-blind process) any kind of external software release must undergo certain institutional approval processes that take a significant amount of time and are out of our control. We ask the referees if the following would be acceptable to address their concerns about reproducibility:
>     * We will include a note in the manuscript saying that the “the code may be made available upon personal request to the authors, pending institutional approval”.
>     * We are happy to further specify, either as an appendix and/or within the main text, any additional details to ensure reproducibility. We remind the referees that our architecture is described in Section 3, and our hyperparameter choices are defined at the beginning of Section 5. However, if the referees have other specific details they’d also like us to include here, please let us know!
>
> * We’d like to point out the discussion and motivation behind the charge matrix in Section 3.2. The main idea is that for a weighted digraph a single charge parameter is not sufficient to incorporate edge weight information, see the second paragraph in 3.2 for the interpretation of a unique charge parameter for each edge in the graph. This seems to be the first attempt to remedy this problem and use a charge matrix in the magnetic Laplacian. For regularization, we saw no evidence of overfitting or time to convergence in the train-test curves with the charge matrix and as such we didn’t explore remedies with additional regularization. However, there is computational cost when introducing another $n^2$ set of parameters, as seen in Table 5. However, the big-O notation of the overall network does not change as the majority of the parameters are still dominated by the learnable matrices $W_{self}^{(l)}$, $W_{neigh}^{(l)}$ and $B^{(l)}$ seen in Equation 8. While we don't yet have proof-theoretical results on this magnetic Laplacian with the charge matrix, one avenue we would like to explore is whether we can predict HMN outperforming other hypergraph neural networks based on various measures of "directionality" or asymmetry that are present in the transition matrix $P$, see response to reviewer 6krg.
>
> * It is an excellent suggestion to elaborate on the justification and potential limitations of degree-based EDVW for citation networks. We’ve added a paragraph elaborating on the interpretation of EDVW in the context of citation networks, and discuss potential limitations and alternatives.
>
> * The hyperparameters were fixed by default according to the recommendations of the authors of the original papers and for our model the hyperparameters were set to the same values. While we can’t provide tuning guidance specific to charge matrix (we emphasize, of course, the charge matrix is a learnable matrix, not a set of hyperparameters), we observed little to no differences in the shape of the train-test curves when using the charge matrix and when using the single charge hyperparameter. In particular, we didn’t see the need for additional regularization despite the addition of more learnable parameters in the matrix introduced by the charge matrix. We didn’t include these curves in the original submission, but if the reviewer believes this would strengthen the paper we can add these plots and a discussion. Some tuning guidance and hyperparameter sets that have been searched over in previous magnetic Laplacian-based neural networks, for example, can be found in Zhang et al., Fiorini et al., and He et al.

---

### Decision · Action_Editor_YGKE · 2025-03-25

**Recommendation:** Accept as is

**Comment:**

As requested by two reviewers, I would ask the authors to try their best to make the code available. The authors argued that this might not be possible due to one author's employment contract.

For future submissions, I would recommend highlighting the changes after the review in a different color to simplify the review process for the reviewers and AC.

**Audience:**

A subset of the graph learning community will be interested in the findings of the present paper. Hence, the paper meets the sufficient conditions for acceptance.

**Claims And Evidence:**

All claims are sufficiently supported. All reviewers endorsed the acceptance of the present work and stated that the requested changes were adequately addressed.